# Systematic approach to school-based assessments for autism spectrum disorders to reduce inequalities: a feasibility study in 10 primary schools

Barry Wright,[1,2] Kalliopi Konstantopoulou ,[3] Kuldeep Sohal,[4] Brian Kelly,[4] Geoff Morgan,[5] Cathy Hulin,[4] Sara Mansoor,[6] Mark Mon-Williams[4,7,8,9]

For numbered affiliations see end of article.

**Correspondence to**
Dr Kalliopi Konstantopoulou; artekon84@gmail.com

## ABSTRACT

**Objectives** This was a pilot study to explore whether the Early Years Foundation Stage Profile (EYFSP) carried out by UK teachers within the 'reception' year, combined with the Social Communication Questionnaire (SCQ), can lead to early identification of children with autism spectrum disorders (ASD) and early access to intervention and can reduce inequity in access to assessment and intervention.

**Design** Pragmatic prospective cohort.

**Setting** Ten primary schools from the SHINE project in Bradford.

**Participants** 587 pupils from 10 schools who transitioned from reception to year 1 in July 2017 and had the EYFSP completed were included in the final study.

**Interventions** The assessment involved a multidisciplinary team of three staff who completed Autism Diagnostic Interview–Revised, Autism Diagnostic Observation Schedule Version 2, classroom observations with an ASD checklist, a teacher-based ASD questionnaire and a final consensus meeting.

**Primary outcome measure** National Institute for Health and Care Excellence guideline-compliant clinical diagnosis of ASD.

**Secondary outcome measures** Age of diagnosis, demographic data and feasibility parameters.

**Results** Children with low scores on the EYFSP were more likely to score above the SCQ threshold of 12, indicating potential autism (50% compared with 19% of children with high scores on the EYFSP (p<0.001)). All children scoring above the SCQ threshold received a full autism assessment; children who scored low on the EYFSP were more likely to be diagnosed with autism (and other developmental issues) compared with those who did not.

**Conclusions** We identified nine new children with a diagnosis of ASD, all from ethnic minorities, suggesting that this process may be addressing the inequalities in early diagnosis found in previous studies. All children who scored above the SCQ threshold required support (ie, had a neurodevelopmental disorder), indicating the EYFSP questionnaire captured 'at-risk' children.

## INTRODUCTION
### What is autism

Autism spectrum disorders (ASD) occur in approximately 1.6% of the UK population.[1]

### Strengths and limitations of this study:

► Consent was sought from all parents regardless of the language by flexible use of interpreters.
► Education and health data were shared, yielding significant benefits.
► We conducted the Social Communication Questionnaire (SCQ) (threshold of 12) with children who scored ≤9 in the Early Years Foundation Stage Profile (EYFSP) and with a random subsample from the high EYFSP group (15% of children with score ≥10).
► All children with a score ≥12 on the SCQ received a detailed comprehensive autism spectrum disorder assessment, and the rest had a teachers' screening questionnaire.
► Any child who had already had a diagnosis on the autism spectrum from the local diagnostic services was also noted.

ASD is a neurodevelopmental condition that often includes a range of repetitive behaviours, preoccupations and interests,[2] and large developmental differences in social communication relative to neurotypically developing individuals.[3] ASD leads to a need for different approaches to education[4] and parenting,[5 6] which can be costly for local authorities[7] and stressful for the parents and the family.[8 9]

### Early identification

Early identification and early intervention have shown promise in improving outcomes.[5 10] Screening young children in early education settings has been attempted, but it captures relatively low numbers of children with ASD[11] despite large numbers (14%) being identified at risk. This has made cost-effective whole population screening problematic,[12] and there is a need for more nuanced approaches. The ability to use routine data to identify 'at-risk' populations

remains the holy grail of autism assessment.[12] The need for such approaches was shown within a large survey of parents in the UK who reported receiving a diagnosis late in primary school despite symptoms being present from infancy.[13] This was confirmed by the Care Quality Commission which reported that children with ASD experience long waits for diagnosis and interventions.[14]

### Early Years Foundation Stage Profile

Recent studies suggest that using the Early Years Foundation Stage Profile (EYFSP)[15] may identify children with a higher risk of ASD.[16] The EYFSP is completed by teachers in England at the end of the reception year and scores 17 different domains of development in terms of whether a child is at an expected level or ahead or behind that level. It is used as a mechanism for flagging children who may need additional help in school and to benchmark UK school profiles.[15]

### Equality of access

Recent work has shown that the diagnosis of autism is less likely to be made early in families from poor backgrounds or from families from ethnic minority groups[17]—reflecting inequalities reported elsewhere.[18] This problem with equity of access could be addressed by having a more widely available process for identifying children with neurodevelopmental disorder as early as possible. One mechanism for improving equity of access is school-based assessments.[19]

### Reasons for feasibility work

To plan a larger study, it is necessary to gather feasibility information for improved assessment processes. We report a feasibility study of a two-stage screening process involving the EYFSP, followed by an established well-validated ASD screening questionnaire—the Social Communication Questionnaire (SCQ).[20] We sought to test the feasibility of a process where children went through this screening process and were then assessed more comprehensively for ASD *in schools* with education and health professionals working together over the course of 1 day.

## METHODOLOGY
### Background

The research was set within the larger Born in Bradford cohort study.[21] We obtained consent from 10 primary schools in an existing consortium, the SHINE partnership. The SHINE group is a group of 10 primary schools that act as a testbed for new approaches to improve services, reduce inequalities and test innovations.[22] We obtained ethical approval from the University of Leeds and Bradford Teaching Hospitals National Health System (NHS) Foundation Trust (IRAS Number: 233328).

### Consent

All parents were approached with a family information leaflet and a consent form. A researcher was available by phone, email or face-to-face for those wishing to discuss the project further. Interpreters were available because many of the population had a first language that was not English.

### Design

Five hundred and ninety-six children in year 5 were available in 10 primary schools, and we approached all those who had received an EYFSP scored by their teachers at the end of the reception year in the summer of 2017.

The study was designed to test feasibility for a larger study.

### Measures

A screening measure to identify children at risk was derived from five items of the EYFSP carried out by teachers at the end of the reception year. The measure was taken from four main symptom areas defined in the research diagnostic criteria for ASD—namely, social reciprocity, language and communication, imagination delays, and repetitive and stereotyped patterns of behaviour. This is described in more detail in a previous study.[16] EYFSP assessment scores are recorded for children in reception who are aged 4–5 years. The assessments conducted by the clinicians occurred in year 1 when children are typically aged 5–6 years. We chose a score threshold of 9 which a previous study found to be significantly (statistically) associated with over 50 times the risk of autism: 52.7 (95% CI: 25.2 to 110.5).[16] Children were dichotomously grouped into 'low' (≤9) and 'high' (≥10) scorers.

The teachers of children with low EYFSP scores and a 15% randomised subgroup of those with high scores (≥10) completed an SCQ,[23] which is a well-established validated autism screening questionnaire with good sensitivity and specificity scores. In previous studies, the SCQ has been found to be helpful in identifying young children with ASD.[24] A threshold score of ≥12 on the SCQ was chosen based on previous research,[25] with claims that this is the best threshold with optimum sensitivity to discriminate between children with and without ASD.[26] A sensitivity analysis was prospectively agreed for the threshold ≥15

## METHODS
### Data linkage allowed us to combine school and health data[26]

All children and families with low EYFSP scores and above threshold SCQ (>12) were offered a National Institute for Health and Care Excellence guideline-compliant ASD assessment, with additional clinical screening assessment for other neurodevelopmental problems including speech and language difficulties, learning difficulties, physical health problems, anxiety and low self-esteem. A 15% randomised subgroup of those scoring high (≥10) in EYFSP had the SCQ completed, and those who scored ≥10 in EYFSP and ≥12 on the SCQ were then also assessed comprehensively in the same way. There were 596 children in the 10 schools; 587 were included in the study as 9

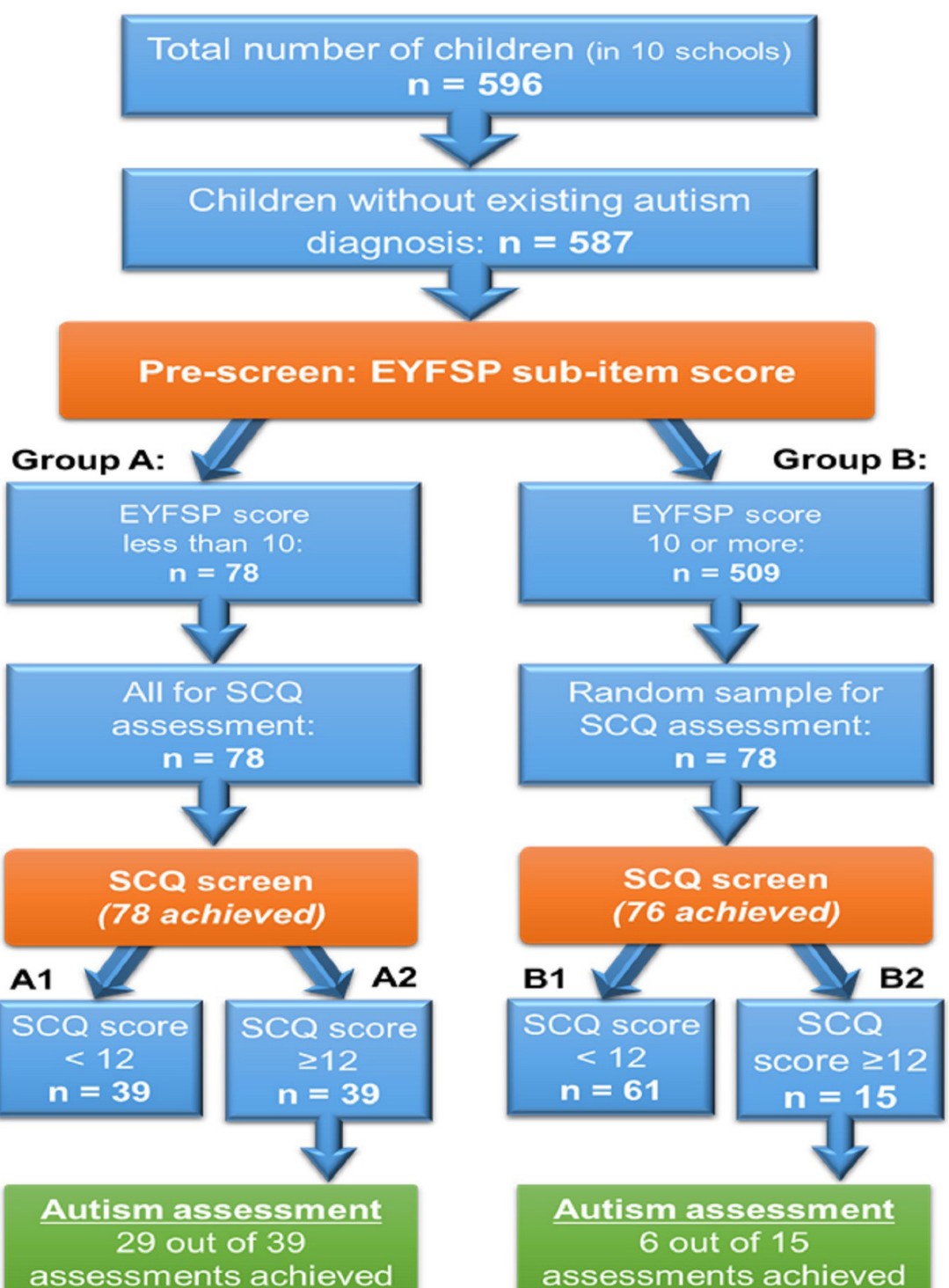

**Figure 1** Number of children who had an autism assessment according to the EYFSP and SCQ scores. EYFSP, Early Years Foundation Stage Profile; SCQ, Social Communication Questionnaire.

children from this cohort had a pre-existing autism diagnosis. Fourteen[14] families decided that they did not want to be part of the study and did not consent. Two families moved to a different school (figure 1).

To check for false negatives, we added an additional screening check for the children in the above groups. In cases where the SCQ was scored below the threshold of 12, teachers filled in a narrative behaviour questionnaire

mapping to the WHO research diagnostic criteria for ASD.[27] This yields a score of 0–12 to identify areas of concern in any of the 12 symptom groups for ASD.[27] Any child who had already had a diagnosis on the autism spectrum from the local diagnostic services was also noted.

Finally, sensitivity analysis was carried out using a cut-off of 15 for the SCQ instead of 12 as this has been used in some studies.[28]

## Patient and public involvement

There has been strong involvement and co-design of this research through the Born in Bradford governors' group, the Connected Yorkshire Patient and Public Involvement panel, SHINE schools, parents, young people and other stakeholders. They have been supportive in the preparatory workshops, feasibility phases and information design of the study. We consulted with the Connected Yorkshire Patient and Public Involvement panel throughout the life cycle of this study who acknowledged the importance to improve the pathway to earlier diagnosis of child ASD to improve children's health and well-being outcomes. The panel consists of parents who have children diagnosed with child ASD or have children who are on the neurodevelopmental disorder care pathways. Some of the discussions focused on the stigma within certain communities in Bradford with certain mental health issues, which results in parents not acknowledging the child's health issues and seeking diagnosis earlier or seeking the appropriate support across health or the education sectors.

We have also extensively engaged with the headteachers at the Bradford SHINE primary schools and other school staff who helped to inform parents of the study and in the recruitment phase. The Bradford SHINE schools were actively involved in the design and implementation phase, and we wish to acknowledge our gratitude in the supporting, co-designing and active involvement in this study.

We disseminated information on the study via the local radio stations including Bradford Ramadan and BBC Radio 4 and via a website to inform individuals of the research that is being undertaken in the region: https://caer.org.uk/autism-spectrum-conditions/.

We have also disseminated the results of the study through dedicated workshops at the Born in Bradford event in September 2019 and through a further workshop in January 2020. These workshops consisted of a broad range of professional stakeholders from health and education across the region that are involved in the care pathway and in public representation. The discussions have evolved to how the research study could be scaled across the region.

## The autism assessment

The assessments took place in the 10 schools in Bradford between September 2018 and July 2019. The assessment involved a team of three multidisciplinary staff drawn from a bank of child and adolescent mental health service clinicians and educational psychologists. The assessment was completed in school in 1 day. One experienced clinician who was trained in Autism Diagnostic Interview–Revised[27] carried out the parent-based semistructured interview with a parent or primary caregiver. Two other professionals (usually an educational psychologist and a clinical psychologist or child psychiatrist) trained in Autism Diagnostic Observation Schedule Version 2[29] carried out the play-based/interaction-based assessment with the child, using the most appropriate module depending on the child's developmental ability and language development. The assessment was carried out by one person and observed by a second with information shared during coding. One of the clinicians also observed the child in class with a bespoke ASD checklist. The clinicians went through a teacher-based questionnaire related to the teacher's experiences of the child's skills and behaviour, including the main symptoms of ASD, using the WHO International Classification of Diseases Version 10 Research Diagnostic Criteria.[30] Finally, there was a consensus meeting with the three external assessors and the teacher, identifying an overall consensus for the presence or absence of definite, possible or no difficulties in the 12 main research diagnostic criteria areas for ASD diagnosis.[28] In the afternoon, each of the clinicians contributed to one single report using a range of subheadings and organised material according to those subheadings. This included a final consensus formulation, a description of strengths and difficulties, and a range of recommendations. As agreed in ethical approvals, the report fell short of making a National Health Service diagnosis (as this was a research project). It was suggested, where appropriate, that referral was made through appropriate local assessment pathways with the report. A range of other recommendations were made, including referral elsewhere (eg, speech and language therapy assessment), physical health checks or a proposed assessment for an Education Health Care Plan, educational psychology assessment or a range of actions. Given the breadth of experience of the assessing professionals and the teacher, a number of possible recommendations for assessment were possible.

## Feasibility outcomes

Feasibility outcomes were collected. These included numbers consenting, attrition rates after consent, acceptability of assessment elements, recording of any language or interpreting issues, and the acceptability and completion of questionnaires.

We conducted qualitative interviews to obtain in-depth information from parents, teachers and clinicians about the acceptability, usefulness and real-world provision of the assessment process.

## RESULTS

Five hundred and ten children scored ≥10 on the EYFSP and 86 children scored ≤9 (at-risk children). Of the 86 children scoring ≤9, 8 (9%) already had a diagnosis on the autism spectrum and the remainder were given the SCQ with threshold results ≥12 and ≥15 reported below[31] (see table 1).

All but one child who met the criteria for a diagnosis of ASD had an SCQ ≥15, meaning that 11 assessments were needed to identify one extra child with ASD.

Of the 510 children who scored ≥10 on the EYFSP (ie, a low risk score), one child already had a diagnosis on the autism spectrum. We conducted the SCQ on a randomised sample (15%) of these children. Seventy-eight families

**Table 1** Percentage of children who met the threshold for ASD with threshold results ≥12 and ≥15 on the SCQ

**SCQ scores (those with score ≥12)**

| ASD | Low EYFSP | Not low EYFSP |
|---|---|---|
| Yes | 9 | 0 |
| No | 20 | 6 |
| Total | 29 | 6 |

31% of those with low EYFSP had a diagnosis of ASD

**SCQ scores (those with score ≥15)**

| ASD | Low EYFSP | Not low EYFSP |
|---|---|---|
| Yes | 8 | 0 |
| No | 13 | 3 |
| Total | 21 | 3 |

38% of those with low EYFSP had a diagnosis of ASD

ASD, autism spectrum disorders; EYFSP, Early Years Foundation Stage Profile; SCQ, Social Communication Questionnaire.

**Table 2** Comparison between EYFSP and SCQ groups

| | SCQ screen | | |
|---|---|---|---|
| EYFSP subscore prescreen | High SCQ | Low SCQ | Total |
| Group A | 50% | 50% | 78 |
| Group B | 19% | 81% | 78 |
| Total | 35% | 65% | 156 |

Pearson's $\chi^2(1)=16.3137$; p<0.001.
Group A are those who score low on the EYFSP subscore prescreen.
Group B are those who do not score low on the EYFSP subscore prescreen.
High SCQ are those who score at least 12 on the SCQ (potential autism).
Low SCQ are those who score less than 12 on the SCQ (no potential autism).
EYFSP, Early Years Foundation Stage Profile; SCQ, Social Communication Questionnaire.

completed the SCQ, with 15 scoring ≥12 on the SCQ, 61 scoring ≤11 and 2 lost to follow-up. The comprehensive autism assessments described were offered to 54 children scoring ≥12 on the SCQ from the children scoring 9 or below on the EYFSP with 39 carried out and with the random subgroup of those scoring 10 or above (n=15). Teachers completed a comprehensive questionnaire based on the WHO research diagnostic criteria for ASD for 20 of 39 children who scored ≤9 in EYFSP and ≤11 in SCQ, as well as for 33 of 61 children who scored ≥10 in EYFSP and ≤11 in SCQ. We received a total of 53 questionnaires, and none of them scored more than 2 out of 12 on the research diagnostic criteria risk checklist, all below the level where a diagnosis of ASD would be likely. The large majority (88.68%) had zero indicators.

Those in group A (who scored low on the EYFSP subscore prescreen) were more likely to be identified as potentially at risk of having ASD on the SCQ screening test compared with those in group B (those who did not score low on the EYFSP subscore prescreen); 50% of those in group A scored ≥12 on the SCQ, compared with 19% in group B (see table 2).

Families of children who scored ≥12 on the SCQ screening tool who were then offered a full autism assessment are described in table 2. Those who scored low on the EYFSP subscore prescreen and then scored high on the SCQ score (indicating potential ASD) were much more likely to be diagnosed with ASD after the full assessment, compared with those in group B (those who did not score low on the EYFSP subscore prescreen and then scored high on the SCQ score). Thirty-one per cent of those in group A with an SCQ ≥12 met the research diagnostic criteria for ASD diagnosis. None of those in group B with an SCQ ≥12 met the research diagnostic criteria for ASD diagnosis.

Tables 3 and 4 indicate the suggested referrals to other services that arose from the assessment, indicating that this process may be useful in identifying children with a range of neurodevelopmental problems and not simply those with ASD.

We checked the general practitioner (GP) records of those 35 children identified as having low (29 children) and not low (6 children) EYFSP scores and ≥12 on the SCQ. Only four of these children had previously had any Read codes recorded for intellectual disability, language delay or disorder, attention deficit hyperactivity disorder or ASD, all four being recorded as having speech delay or disorder of speech and language. Two of these four children were assessed in our study as meeting the criteria for ASD. The remaining 31 children with low and not low EYFSP and SCQ >12 had no GP-recorded Read codes, but all 31 had additional needs that were newly identified in our assessments (see table 4). This shows that of the 35 children, 31 would gain new interventions as a result of our assessment processes that they were not currently accessing. All nine of the children who were newly diagnosed with ASD by this research were from an ethnic minority background. There were six boys and three girls who were diagnosed with ASD. Of the six boys, three were of Pakistani origin, two of Bangladeshi origin and one of gypsy/traveller origin. Of the three girls who were diagnosed with ASD, two were of Pakistani origin and one of Bangladeshi heritage.

### Qualitative findings

Associated qualitative research will be published separately. Feedback was requested from clinicians, school staff, assessed children's parents and parents of children with a neurodevelopmental disorder from a patients' panel.

Both parents and clinicians were positive about school-based assessment occurring (largely) in 1 day. This included the benefits of the child being in their normal routine and experiencing less anxiety than clinic visits. Parents were positive about not having to chase appointments, and teachers were positive about involvement in all assessments.

**Table 3** Assessment outcomes according to risk groups for children scoring at least 12 on the SCQ (potential autism)

| | Group A2 | Group B2 | Groups A2 and B2 |
|---|---|---|---|
| **Referral to service** | **Prescreen: low EYFSP subscore (n=29)** | **Prescreen: not low EYFSP subscore (n=6)** | **Total with autism assessment (n=35)** |
| Autism spectrum disorder | 9 (31.0%) | 0 (0%) | 9 (25.7%) |
| Assessed need for external (outside school system) support | 22 (75.9%) | 3 (50.0%) | 25 (71.4%) |
| Assessed need for internal (within school system) support | 29 (100%) | 5 (83.3%) | 34 (97.1%) |
| Assessed need for internal or external support | 29 (100%) | 6 (100%) | 35 (100%) |

Group A2 are those scoring low on the EYFSP subscore prescreen and scoring at least 12 on the SCQ (potential autism).
Group B2 are those not scoring low on the EYFSP subscore prescreen and scoring at least 12 on the SCQ (potential autism).
EYFSP, Early Years Foundation Stage Profile; SCQ, Social Communication Questionnaire.

Clinicians valued multidisciplinary working and the positives of access to rich school-based data. A special educational needs co-ordinator (SENCO) from one of the school mentioned that 'I liked that everybody can come together because you are in one place, everybody that knows the child is there and then it is kind of written as a team around the child …' Parents commented that including school in the assessment process had helped teaching staff to adapt teaching and support for the child promptly. Challenges identified included difficulties coordinating different professionals, children and parents together and last-minute cancellations: 'this process was highly dependent on administration

**Table 4** Recommendations from assessing clinicians about additional support needed for 35 assessed children

| | Group A2 | Group B2 | Group A2 and B2 |
|---|---|---|---|
| **Enacted onward referral to service** | **Prescreen: low EYFSP subscore (n=29)** | **Prescreen: not low EYFSP subscore (n=6)** | **Total with autism assessment (n=35)** |
| Autism spectrum disorder | 9 (31.0%) | 0 (0%) | 9 (25.7%) |
| Speech and language therapy assessment | 16 (55.2%) | 3 (50.0%) | 19 (54.3%) |
| Nurture group/encouragement of social interaction/monitoring | 12 (41.4%) | 4 (66.7%) | 16 (45.7%) |
| Learning needs assessment | 4 (13.8%) | 2 (33.3%) | 6 (17.1%) |
| In-school LEGO-based therapy | 3 (10.3%) | 0 (0%) | 3 (8.6%) |
| Parent support | 3 (10.3%) | 0 (0%) | 3 (8.6%) |
| Dyslexia assessment | 3 (10.3%) | 0 (0%) | 3 (8.6%) |
| Dyscalculia assessment/maths skills support | 1 (3.4%) | 0 (0%) | 1 (2.9%) |
| Educational psychology/cognitive assessment | 9 (31.0%) | 0 (0%) | 9 (25.7%) |
| Formal EHCP triggered | 5 (17.2%) | 0 (0%) | 5 (14.3%) |
| Visual aids and/or vision assessment | 5 (17.2%) | 0 (0%) | 5 (14.3%) |
| In-school creative activities groups | 3 (10.3%) | 0 (0%) | 3 (8.6%) |
| Gross motor skills support | 3 (10.3%) | 1 (16.7%) | 4 (11.4%) |
| Physical health check | 2 (6.9%) | 0 (0%) | 2 (5.7%) |
| In-school social story intervention | 2 (6.9%) | 0 (0%) | 2 (5.7%) |
| New adaptations in classrooms | 6 (20.7%) | 0 (0%) | 6 (17.1%) |
| Occupational therapy assessment | 1 (3.4%) | 0 (0%) | 1 (2.9%) |
| Other group support | 1 (3.4%) | 0 (0%) | 1 (2.9%) |
| Attention concentration support | 6 (20.7%) | 1 (16.7%) | 7 (20.0%) |

Group A2 are those scoring low on the EYFSP subscore prescreen and scoring at least 12 on the SCQ (potential autism).
Group B2 are those not scoring low on the EYFSP subscore prescreen and scoring at least 12 on the SCQ (potential autism).
EHCP, Education and Health Care Plan; EYFSP, Early Years Foundation Stage Profile; LEGO, Leg Godt; SCQ, Social Communication Questionnaire.

both from the project and from school…'. Other themes highlighted related to the diagnosis and a range of responses relating to concern from a parent that their child's problems may be minimised or that they might be stigmatised.

## DISCUSSION

This study has shown that it is feasible to carry out a larger study of a new assessment care pathway for neurodevelopmental problems across a district. We found that schools were very willing to take part in the study and showed great interest in early identification of children with autism and other support needs. All schools we approached in Bradford agreed to take part and facilitate the study. Teachers were supportive, completing 53 of 55 questionnaires about the children who did not receive the full autism assessment. The acceptability to families is relatively good, although some families withdrew from the study and some had concerns about the consequences of their child receiving a diagnosis of ASD. This suggests that care needs to be taken when considering the emotional consequences for the family. It is good practice to provide parenting support to families of children newly diagnosed with ASD, and this should be a key part of new assessment pathways or future research.

In our trial, the EYFSP prescreen identified 13% of the pupil population (78 pupils scoring less than 10 on the EYFSP out of 587 pupils). From this population, half scored highly on the SCQ such that approximately 6.5% of the population received an autism identification with the addition of the EYFSP prescreen. This compares with 14%[11] in similar early life screening studies without a prescreen stage. This has potential cost-effective benefits that we were unable to test but should be key parts of future research.

A recent paper[32] suggests an SCQ threshold of 12, with a sensitivity of 42% and specificity of 89%. Other authors have used 15.[31] Our analysis shows 35 assessments identify nine children with ASD and 23 assessments identify eight children, suggesting cost-effectiveness analysis would be helpful in a larger study. While we cannot accurately assess sensitivity in our study (as we have not assessed all the children in the sample for ASD), we used teacher-based questionnaires (with ASD research diagnostic criteria) in 33 children with normal EYFSP scores and low SCQ scores, and none had more than two flagged areas of concern on the research diagnostic criteria symptom list for ASD (5–6 is the threshold for diagnosis). This suggests that further research may reveal an improved sensitivity when EYFSP is used as a prescreen before SCQ.

This study has shown that there may be promising alternatives to existing assessment pathways for ASD (ie, the use of EYFSP subscore as a prescreen tool before SCQ screening). Advantages to the clinical process include the fact that information can be gathered from the school with those who know the child best (parents/carers and teacher) in 1 day in an environment known to the child, which may give a more accurate assessment. Previous studies using screening instruments with similar sample sizes have found a third of the sample are lost to follow-up.[11] Our study has vastly lower attrition because of the close link with the clinical teams in schools where parents are in regular contact. The early identification of ASD means that children can access the best educational placement early and allows the local authority to plan its services and resources. It may resolve inequalities seen in previous studies where sections of the population do not come forward for assessment.[17 18]

This study identified a number of new children (n=9) with a diagnosis of ASD. This has enabled support to be established early. All these children were from ethnic minorities, suggesting that this process may be addressing inequalities in early diagnosis found in previous studies,[17] although this would need further large-scale research to confirm. In other studies using the SCQ, when children score above the threshold but do not have ASD, approximately 90% have a neurodevelopmental disorder or developmental problem of some sort, requiring identification and support.[33] In our study (using the EYFSP), this was 100% with all children having identified support needs.

The study was limited by its size, suggesting further larger district-level research with cost-effectiveness analysis needs to take place.

**Author affiliations**
[1]Hull York medical School and Department of Health Sciences, University of York, York, UK
[2]Leeds and York Partnership NHS Foundation Trust, Leeds, UK
[3]Child and Adolescent Psychiatry, Leeds Community Healthcare NHS Trust, Leeds, UK
[4]Born in Bradford Research Team, Bradford Institute for Health Research, Bradford Royal Infirmary, Bradford, UK
[5]Educational Psychology Team, City of Bradford, Metropolitan District Council, Bradford, UK
[6]Child and Adolescent Psychiatry, Bradford District Care NHS Foundation Trust, Saltaire, UK
[7]Department of Psychology, University of Leeds, Leeds, UK
[8]Centre for Applied Education Research, Wolfson Centre for Applied Health Research, Bradford Royal Infirmary, West Yorkshire, UK
[9]National Centre for Optics, Vision and Eye Care, University of South-Eastern Norway, Lærerskoleveien, Norway

**Acknowledgements** Thanks to all the professionals who helped with the assessments, including Dr Claudia Salt, Dr Emily Williams, Natalie Langley, Halimah Hafiz, Nabihah Kauser, Ronnie Hartley, Prakash Thapa and Dr Alice Lambert. Thanks also to a range of people for their help to make this research possible, including Professor John Wright, Dr David Sims, Rachael Vann, Sarah Oates, Amy Hart, Sophie Tully, Sarah Gates, Shelley Russell, Catarina Teige, Dani Varley, Dr Sue Lee, Rebecca Joy, Misbah Khan and Lydia Phillip. A special thanks to Dr Stefan Williams and Dr Sujo Anathhanam who carried out earlier preparatory work prior to this study. We would like to acknowledge our gratitude to the Bradford SHINE schools, Born in Bradford governors' group, Connected Yorkshire Patient and Public Involvement panel, parents and their children, and other stakeholders who have been involved in the study.

**Contributors** BW conceived the presented idea, and contributed to the design and delivery of the project and the writing of the manuscript. KK contributed to the

design, delivery, data collection and the writing of the manuscript. KS contributed to the design of the project and agreed with the manuscript's results and conclusions. BK contributed to the design of the project, completed the statistical analysis of the project and contributed to the writing of the manuscript. GM contributed to the design and delivery of the project and agreed with the manuscript's results and conclusions. CH contributed to the design, overall organisation, data collection and writing of the manuscript. SM contributed to the design and delivery of the project. MM-W contributed to the design of the project and agreed with the manuscript's results and conclusions.

**Funding** The work was conducted within infrastructure provided by the Centre for Applied Education Research (www.caer.org.uk), and funded by the Department for Education through the Bradford Opportunity Area. The views expressed are those of the author(s), and not necessarily those of the National Health System (NHS), the Bradford Local Authority or the Department for Education.M. Mon-Williams was supported by a Fellowship from the Alan Turing Institute. The work was conducted within infrastructure provided by the Centre for Applied Education Research (funded by the Department for Education through the Bradford Opportunity Area) and ActEarly: a City Collaboratory approach to early promotion of good health and wellbeing funded by the Medical Research Council (grant reference MR/S037527/). M. Mon-Williams involvement was supported by the National Institute for Health Research Yorkshire and Humber ARC (reference: NIHR20016). The views expressed in this publication are those of the author(s) and not necessarily those of the National Institute for Health Research or the Departments of Health and Social Care or Education.

**Competing interests** None declared.

**Patient consent for publication** Not required.

**Provenance and peer review** Not commissioned; externally peer reviewed.

**Data availability statement** Data are available upon reasonable request.

**ORCID iD**
Kalliopi Konstantopoulou http://orcid.org/0000-0001-5606-0481

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
