## [Reviewer comments · BMJ Open]

ARTICLE DETAILS

TITLE (PROVISIONAL)	A systematic approach to school based assessments for Autism Spectrum Disorders to reduce inequalities: A feasibility study in ten primary schools.
AUTHORS	Wright, Barry; Konstantopoulou, Kalliopi; Sohal, Kuldeep; Kelly, Brian; Morgan, Geoff; Hulin, Cathy; Mansoor, Sara; Mon-Williams, Mark

VERSION 1 – REVIEW

REVIEWER	Josefa Canals-Sans Rovira i Virgili University Spain
REVIEW RETURNED	20-Jul-2020

GENERAL COMMENTS	This paper has an innovative objective with clinical and educational relevance. Results indicate that there may be promising alternatives to ASD existing assessment. The research has a good design and the methods are adequate. However, I have the following suggestions to be considered: - Methods: More information on age and sex, and other sociodemographic data of the families are needed. Although in Discussion more detection of ASD in ethnic minorities is referred, this data is not presented in Methods nor Results. It would be interesting to know if children with ASD risk and without ASD previous diagnosis were of groups with social disadvantages or how many girls were detected and / or diagnosed.- Pg 7: In the following sentence: "All those children and families with low EYFSP scores and above threshold SCQ (>12) were offered a NICE guideline compliant ASD Assessment, with additional clinical screening assessment for other developmental problems", more information on other problems is needed.- Pg 8.: The study of the sensitivity and specificity of cut-off of SCQ would be the objective of another study. The authors have justified the score of 12 but in parallel to the goals of this study they offer data using a cut of 15 or the SCQ instead of 12. Also, the data of comparison between 12 and 15 obtained in this study are not discussed. I suggest that it be removed from this article.- Pg 9, 10, 11: Information on sample and referred in Figure is repeated. I recommend being concise with information.-Pg11: Table2: The title must include "comparison between EYFSP and SCQ groups"-Pg 12. Why this format in these two sentences?• 31% of those in group A with a SCQ of 12 or more met the research diagnostic criteria for ASD diagnosis.• None of those in group B with a SCQ of 12 or more met the research diagnostic
---

	criteria for ASD diagnosis. Pg 12, Table 3: I suggest changing the title to Assessment outcomes according to risk groups. Also, explanation on Groups A2 and B2 must be included in Table . Pg 13, Table 4: The group A also include children with SCQ>12 apart from "Pre-screen:Low EYFSP sub-score, n=29" Pg 14. It is difficult to understand the reason there exist this part: "Feasibility outcomes". It is not understood where these figures come from now -Discussion: Overall it's a bit poor. More information to compare other screening alternatives and early diagnosis should be considered.
--	--

REVIEWER	Mats Cederlund Gothenburg University Sweden
REVIEW RETURNED	22-Jul-2020

GENERAL COMMENTS	Dear authors, Thank you for an interesting paper that highlights at least two important issues. Firstly, how to be able to get the most complete assessment possible of a child with neurodevelopmental/neuropsychiatric problems, which would be easier working side by side with the respective child's school the way you propose here. Secondly, how to find children coming from another language background in need of neurodevelopmental/neuropsychiatric assessment. With this study you have been able to address these issues! I believe that you should give the manuscript some work. There is sometimes a lack of "flow" in the manuscript that makes it difficult to follow. Words are left out in some places. Generally it is advisable not to start a sentence with a figure, albeit with the number written out in text instead (e.g. Fifty instead of 50). Kind regards, The reviewer
--

VERSION 1 – AUTHOR RESPONSE

A systematic approach to school based assessments for Autism Spectrum Disorders to reduce inequalities? A feasibility study in ten primary schools		
Comment No.	Comment	Response
Reviewer 1 – Josefa Canals-Sans – Rovira I Virgili University, Spain.		
1	This paper has an innovative objective with clinical and educational relevance. Results indicate that there may be promising alternatives to ASD existing assessment.	We are grateful to the reviewer for their positive comments. We have responded to the suggestions (which

	The research has a good design and the methods are adequate. However, I have the following suggestions to be considered:	we hope are useful and informative) below.
2	Methods: More information on age and sex, and other socio-demographic data of the families are needed. Although in Discussion more detection of ASD in ethnic minorities is referred, this data is not presented in Methods or Results. It would be interesting to know if children with ASD risk and without ASD previous diagnosis were of groups with social disadvantages or how many girls were detected and / or diagnosed.	We have provided more information as requested. The study did not set out to establish population norms (though we have addressed this issue and relating inequalities in other published work – as referenced within the revised manuscript) so whilst these children were from ethnic minorities we can't speak to the broader issue of ethnicity per se. We now report the ethnicity and sex of the nine children diagnosed with ASD. There were six boys and three girls diagnosed with ASD. From the six boys, there are three of Pakistani origin, two of Bangladeshi origin and 1 gypsy/traveller origin. From the three girls diagnosed with ASD, two are of Pakistani origin and one is Bangladeshi. EYFS assessment scores are recorded for children in Reception who are aged from 4 to 5 years of age. The assessments conducted by the clinicians occurred in Year 1 when children are aged 5 to 6 years of age. We have updated the document to reference these ages.
3	Pg. 7: In the following sentence: "All those children and families with low EYFSP scores and above threshold SCQ (>12) were offered a NICE guideline compliant ASD Assessment, with additional clinical screening assessment for other	Other problems include speech and language problems, learning

	developmental problems”, more information on other problems is needed.	difficulties, physical health problems, anxiety and low self-esteem. We have updated the document to report these other neurodevelopmental problems.
4	Pg. 8: The study of the sensitivity and specificity of cut-off of SCQ would be the objective of another study. The authors have justified the score of 12 but in parallel to the goals of this study they offer data using a cut of 15 or the SCQ instead of 12. Also, the data of comparison between 12 and 15 obtained in this study are not discussed. I suggest that it be removed from this article.	We have kept the thresholds of 12 and 15 as they have both been proposed by previous authors (referenced in the text). We believe this to be important as an additional 11 assessments needed to take place to identify one child with ASD when using the threshold of 12 instead of 15. This becomes important feasibility information therefore and we have as suggested mentioned this in the discussion.
5	Pg. 9, 10, 11: Information on sample and referred in Figure is repeated. I recommend being concise with information.	We have amended the methods section where figure 1 is presented with concise information on the sample. This information which was in the first paragraph of the results section has now been removed.
6	Pg.11: Table2: The title must include “comparison between EYFSP and SCQ groups	This has been updated.
7	Pg. 12, Why this format in these two sentences?  • 31% of those in group A with a SCQ of 12 or more met the research diagnostic criteria for ASD diagnosis. • None of those in group B with a SCQ of 12 or more met the research diagnostic criteria for ASD diagnosis. 	This has been updated so that it is in the same format.
8	Pg. 12, Table 3: I suggest changing the title to Assessment outcomes according to risk groups. Also, explanation on Groups A2 and B2 must be included in Table.	We have updated the title of Table 3. Groups A2 is a subset of Groups A – i.e. those with a score on the SCQ of 12 or more; and similarly with group B2.

		We have added additional text to the explanation. So it becomes: “Group A2 are those scoring low on the EYFSP sub-score pre-screen score and scoring at least 12 on SCQ (potential autism)” “Group B2 are those not scoring low on the EYFSP sub-score pre-screen score and scoring at least 12 on SCQ (potential autism)”
9	Pg. 13, Table 4: The group A also include children with SCQ>12 apart from “Pre-screen: Low EYFSP sub-score, n=29”	The groups in Table 4 should be the same as Table 3 – that is Groups A2 and B2 (rather than Group A and Group B) – so have changed this in the text – also would need the same description of the groups as was requested for Table 3 in the above comment – have added this to the text as well.
10	Pg. 14, It is difficult to understand the reason there exist this part: “Feasibility outcomes”. It is not understood where these figures come from now	We have removed this section and have added points on feasibility into the discussion. “We found that schools were very willing to take part in the study, and showed great interest in early identification of children with autism and other support needs. All schools we approached in Bradford agreed to take part and facilitate the study. Teachers were supportive, completing 53 of 55 questionnaires about the children who did not receive the full autism assessment.”
11	Discussion: Overall it's a bit poor. More information to compare other screening alternatives and early diagnosis should be considered.	We have tried to improve the discussion section in the revised version.

Comment No.	Comment	Response
Reviewer 2 – Mats Cederlund – Gothenburg University, Sweden		
1	An interesting paper that highlights at least two important issues. Firstly, how to be able to get the most complete assessment possible of a child with neurodevelopmental/neuropsychiatric problems, which would be easier working side by side with the respective child's school the way you propose here. Secondly, how to find children coming from another language background in need of neurodevelopmental/neuropsychiatric assessment. With this study you have been able to address these issues!	Our intention was always to give parents the opportunity to have their children assessed in a clinic or in the school. However, we found that the school based approach enabled us to engage with the teachers on the children in the development of the final reports. Schools were very helpful in identifying suitable translators needed for the assessments.
2	There is sometimes a lack of "flow" in the manuscript that makes it difficult to follow. Words are left out in some places.	We have made changes to the manuscript and it has been reviewed by the study team.
3	Generally it is advisable not to start a sentence with a figure, albeit with the number written out in text instead (e.g. Fifty instead of 50).	We have tried to address this issue within the revised manuscript.